# Effect of Metal-Cored Filler Wire on Surface Morphology and Micro-Hardness of Regulated Metal Deposition Welded ASTM A387-Gr.11-Cl.2 Steel Plates

**DOI:** 10.3390/ma15196661

**Published:** 2022-09-26

**Authors:** Din Bandhu, Faramarz Djavanroodi, G. Shaikshavali, Jay J. Vora, Kumar Abhishek, Ashish Thakur, Soni Kumari, Kuldeep K. Saxena, Mahmoud Ebrahimi, Shokouh Attarilar

**Affiliations:** 1Department of Mechanical Engineering, Indian Institute of Information Technology Design and Manufacturing, Kurnool 518008, Andhra Pradesh, India; 2Department of Mechanical Engineering, College of Engineering, Prince Mohammad Bin Fahd University, Al Khobar 31952, Saudi Arabia or; 3Department of Mechanical Engineering, Imperial College London, London SW7 2AZ, UK; 4Department of Mechanical Engineering, G. Pulla Reddy Engineering College (Autonomous), Kurnool 518007, Andhra Pradesh, India; 5Department of Mechanical Engineering, School of Technology, Pandit Deendayal Energy University (PDEU), Gandhinagar 382421, Gujarat, India; 6Department of Mechanical & Aerospace Engineering, Institute of Infrastructure, Technology, Research and Management (IITRAM), Ahmedabad 380026, Gujarat, India; 7Department of Metallurgical and Materials Engineering, Visvesvaraya National Institute of Technology (VNIT), Nagpur 440010, Maharashtra, India; 8Department of Mechanical Engineering, GLA University, Mathura 281406, Uttar Pradesh, India; 9National Engineering Research Center of Light Alloy Net Forming and Key State Laboratory of Metal Matrix Composites, School of Materials Science and Engineering, Shanghai Jiao Tong University, Shanghai 200240, China

**Keywords:** ASTM A387, filler wire, metal-cored, RMD, welding

## Abstract

Environmental and human-friendly welding is the need of the hour. In this context, this study explores the application of the regulated metal deposition (RMD) technique for ASTM A387-Gr.11-Cl.2 steel plates. To examine the effect of metal-cored filler wire (MCFW), MEGAFIL 237 M was employed during regulated metal deposition (RMD) welding of 6 mm thick ASTM A387-Gr.11-Cl.2 steel plates. The welding was carried out at an optimized current (A) of 100 A, voltage (V) of 13 V, and gas flow rate (GFR) of 21 L/min. Thereafter, the as-welded plates were examined for morphological changes using optical microscopy. Additionally, the micro-hardness of the as-welded plates was measured to make corroboration with the obtained surface morphologies. In addition to this, the as-welded plates were subjected to heat treatment followed by surface morphology and micro-hardness examination. A comparison was made between the as-welded and heat-treated plates for their obtained surface morphologies and microhardness values. During this, it was observed that the weld zone of as-welded plates has a dendritic surface morphology which is very common in fusion-based welding. Similarly, the weld zone of heat-treated plates has a finer and erratic arrangement of martensite. Moreover, the obtained surface morphologies in the weld zone of as-welded and heat-treated plates have been justified by their respective hardness values of 1588.6 HV and 227.3 HV.

## 1. Introduction

Welding is one of the most prevalent technical processes, having widespread use in the assembly of metallic parts for autos, in aviation, construction, military, and energy industries [1,2,3,4]. Climate change and the need for sustainable production, on the other hand, are major problems for many manufacturers. In this context, Miller Electric Mfg. introduced a novel welding technique in 2004, known as RMD welding. This technology is typically utilized for quality welding in the petrochemical industries. The discharge of liquid droplets from the filler wire to the molten pool is determined and optimized by modifying the voltage and current frequencies. A particularly designed inverter-based power generator with revolutionary microcontrollers based on a set of programs tunes voltage and current frequencies. The RMD technique offers several improvements over standard welding methods that may substantially improve the weld quality, consistency, and growth of a company. It allows for consistent droplet dispersion. As a consequence, the technician has a less difficult time coping with the molten pool. It solves the concern of low-high mismatch across pipelines by easily bridging distances close to 4.8 mm. It eradicates the necessity of backing gas throughout the cycle, yielding increased production efficiency and decreased weld expenditures. The method seeks to establish a constant weld arc regardless of wire stick-out, allowing rookie technicians to comprehend with little instruction. It is suitable for root pass weld joint since it yields a 3.2–6.4 mm deep throat, alleviating the requisite for a GTAW hot pass. Its travel rates are 6–12 inches/minute, which is faster than GTAW (3–5 inches/minute) and classical welding (3–8 inches/min). As a consequence, the depositing rate increases dramatically. Due to the stable molten pool and consistent metal flow, the method produces slag and spatter-free joints. Splashing against the sides is also reduced. Hence, the methodology alleviates the possibility of reworking over the finished welds, culminating in higher efficiency and quality yet minimizing costs and time [5,6,7,8,9].

In addition to this, the selection of consumables, such as filler wire, is crucial. It is not only for successful and cost-effective manufacture but also for the release of hazardous gases throughout the process. Manufacturers often utilize three types of consumables in arc welding processes: solid filler wire, flux-cored filler wire, and metal-cored filler wire. In 1948, solid wires made their debut. They are predominantly utilized in the welding of sheet metals. They have excellent efficiency and create aesthetic welds, but they have several downsides, such as excessive spatter, poor rate of deposition, slow travel speed, and post-weld cleaning work [10,11,12]. Eventually, in 1957, a brand-new form of filler wire—the flux-cored filler wire (FCFW) was unveiled. They generate clean and strong welds with little spatter. They outperform solid wires in terms of smooth arc generation, penetration depth, and deposition rate; they are also more user-friendly. While FCFW offers some advantages, it also has a few disadvantages. These wires can occasionally cause fissures or slag inclusion in the weldments, resulting in incomplete fusion of sections to be joined. If the fumes, released during the filler wire combustion, do not depart before the weld metal hardens, the welding region is likely to develop pores and turn permeable. In addition, these filler wires are restricted to horizontal and straight placements and are not ideal for thinner sections [11,12,13,14].

Fabricators, on the other hand, were still striving for a consistent approach that might be superior, quicker, and cost-effective. Despite the fact that their search was poised to halt at some point with the creation of FCFW, they were the only ones who could achieve a considerable deposition rate. Moreover, the solid wires were yet to reach significant deposition efficiency. However, in 1973, a novel invention was created in the field of filler wire, i.e., metal-cored filler wire (MCFW). MCFWs are relatively new to the welding industry and are more helpful in terms of quality and productivity, as well as being ecologically benign. These filler wires are supplied in diameters in the range of 0.9–2.4 mm and employ Ar (>75%) + CO_2_ shielding gas combination. [15,16,17,18,19]. MCFW brings together the strengths of solid and flux-cored filler wires. It combines the fast deposition speed of a flux-cored filler wire with the superior performance of a solid filler wire. The deposition rates of all three filler wires have been compared as shown in Figure 1. A 1.2 mm (0.045″) diameter metal-cored filler wire deposits at a rate of 5.4–6.4 kg/h, while a solid filler wire with similar dimension deposits at a rate of 3.6–4.5 kg/h [20].

Quicker travel velocity, relatively high duty cycles, minimal mill scale concerns, spatter-free and slag-free welding, a significant decline in weld imperfections namely porosity, undercut, fusion inadequacy, eradication of clean-up task, and post-weld activities such as grinding are all advantages of metal-cored filler wire [10]. In a study [20], when compared to old-fashioned filler wires, metal-cored filler wires have a lower negative impact on technicians and the environment due to lower amounts of fume production (Figure 2).

The study of surface morphology is essential for understanding the material’s behavior. Therefore, Alipooramirabad et al. [21] applied SMAW, MSAW, and FCAW on HSLA steel. The effect of residual stresses in conjunction with surface morphology and mechanical attributes was examined in this study. The work reveals that the combination of FCAW and MSAW yields high residual stress which further characterizes the presence of bainitic and Widmanstätten ferrite in the material. Zhang et al. [22] welded EH36 and 316 L steels by means of laser arc welding and examined the relation between phase composition and grain distribution in different zones of the weldments. The study reveals the presence of martensitic transformation in the laser zone. Stanisz et al. [23] conducted a comparative study by performing different welding operations on 1.5 mm thick steel. The study examined the macrostructure, surface morphology, and microhardness of the weldments and it was observed that the narrowest zone of hardness was obtained with laser welding whereas the widest hardness zone was obtained with robot-assisted CMT welding. Xi et al. [24] executed laser welding in association with TIG welding to investigate the influence of Fe10Cr4AlRE alloy cracks overlayed on 316 L steel. The outcome of this study recommends this process be utilized in the production of the primary container of CiADS stating the process’s cost-effectiveness and viable approach to the traditional methods of producing the corrosion-resistant FeCrAl alloy. Sepe et al. [25] performed laser welding on AlMg3.5 Mn and AlSi1MgMn T6 and evaluated the performance of tube-tube welded samples based on pure axial load, pure torsion load, constant and variable amplitude load. The welded samples were also subjected to PWHT to examine the impact of residual stresses on the fatigue behavior of the material. The finite element method (FEM) was employed for numerical analyses to estimate the residual stress in the samples. A good agreement was found between the numerical results and the experimental works. Sim et al. [26] welded duplex stainless steel with shielded metal arc welding and investigated the process’s influence on the material after heat treatment. An increase in post-weld heat treatment temperatures was associated with an increase in the austenite phase. The weld metal’s hardness rose with heat treatment temperature but stayed minimum at solid solution annealing temperatures. Sepe et al. [27] analyzed the effect of residual stresses induced by the shielded metal arc welding (SMAW) process during the preparation of dissimilar T-joint using 5 mm thick AISI304 and S275JR steels. Temperature variations during the welding process were examined by incorporating thermocouples. A simulation was accomplished on a finite element model through the “element birth and death” method. Experimental work and numerical model were compared for temperature distribution and distortion and it was found that the finite element model provides a high level of accuracy. Chen et al. [28] performed laser welding on HSLA steel having 800 MPa of yield strength and examined the consequences of heat input on surface morphology and mechanical attributes of the weldment. Martensite and austenite were detected in the fusion zone. The study further reports the excellent tensile strength of the weldment was achieved with lower heat input and therefore recommends the implementation of low heat input during laser welding of HSLA steel. Sepe et al. [29] prepared two-pass V-groove butt welds to examine the impact of thermo-mechanical material characteristics on structural response. The materials properties considered for this study were specific heat, thermal conductivity, thermal expansion coefficient, and young’s modulus. The performance characteristics considered in this work were temperature, angular distortion, displacement, and residual stress. A FEM model was also prepared and compared with the experimental work and a good agreement was found between them.

According to the published studies, several scholars opted to investigate the impact of welding factors on bead geometry utilizing various welding procedures and consumable wires. Nevertheless, the effect of the RMD methodology employing MCFW on various classes of metals has not been thoroughly investigated. Furthermore, few scholars have optimized the welding operation prior to deploying it in practical uses, yet simply limited endeavors have been undertaken to utilize the RMD technique for low alloy steel. The scholars of this study [30,31,32] conducted a number of investigations with a viewpoint of determining the ideal welding conditions for low alloy steel concerning the bead geometry. Current (A) = 100 A, voltage (V) = 13 V, and gas flow rate (GFR) = 21 L/min were the optimal values. This study is a continuation of prior efforts.

Here, the as-welded RMD weldments have been examined for surface morphology and microhardness and compared to those of heat-treated RMD weldments for the same material. The scholars are positive that the present research will contribute significantly to upcoming studies investigating the use of MCFW for sophisticated welding methods.

## 2. Materials and Methods

### 2.1. Experimentation

Two ASTM A387-Gr.11–Cl.2 steel plates of 24 mm × 7.5 mm × 6 mm were welded up through RMD welding (Figure 3c). A ‘MEGAFIL 237 M’ wire (1.2 mm in diameter) was applied as a filler wire. The elemental constitutions of ASTM A387-Gr.11–Cl.2 plates and filler wire are provided in Table 1.

Before welding, the sections were wiped with an iron brush. C_3_H_6_O was next utilized to clean the workpiece’s surfaces of any leftover impurities and O_2_ deposits. Grooving was conducted on both plates (Figure 3a). Subsequently, tacking was performed among both workpieces at a spacing of 03 mm (Figure 3b). These workpieces were then fastened for executing welding operation by means of the “Continuum 500” welding machine at obtained optimal settings [30,31,32]. The particulars of the welding settings are presented in Table 2.

Post-RMD operation, a 10 mm wide section was excised from one side for as-welded microhardness and surface morphology examination. Thereafter, the welded plates were subjected to a heat treatment cycle.

### 2.2. Post-Weld Heat Treatment

In this work, the PWHT has been executed in conformity with the ASME Section-2C and American Petroleum Institute’s approved guideline 934-C [33,34].

For PWHT, the weld samples had been housed in an electric furnace. The PWHT cycle had a temperature variation of 300–700 °C with a heating and cooling speed of 100 °C/h. The heating process begins at 300 °C and progresses at a pace of 100 °C/h. The samples had been submerged in the furnace for 6 h when the temperature approached 700 °C. The cooling process then begins with a cooling speed of 100 °C/h. The hot samples had been removed from the furnace upon approaching 300 °C and treated to atmospheric cooling. The thermal cycle of the heat treatment is shown in Figure 4.

ASTM A387-Gr.11-Cl.2 is a chrome-moly steel and it is hardened; consequently, PWHT is mandatory. These steels are tough and strong enough to endure the internal stress experienced throughout the operation. They also have good oxidation and creep resistance qualities. As a result, they are widely used in the design of high-temperature tubular sections namely superheaters, pressure vessels, compressors, and turbine propellers. The welding of these elements might result in substantial residual stress and morphology alterations. Moreover, it is commonly accepted that the structural qualities of weldment are inferior to those of the base material. Welding-instigated morphological transformations are assumed to be the cause of differences in the structural qualities of the welded metal [35,36,37]. According to a few pieces of research [38,39], a substantial percentage of low alloy steel inadequacies appear in the HAZ zone. PWHT is required to alleviate brittle failure caused by high-stress loading rates in low alloy steel weldments. PWHT lessens residual stress and promotes toughness. It also strengthens dimensional stability and corrosion resistance, as well as lowers hardness in the weld zone [40,41,42].

Post-heat treatment, a further 10 mm wide section was excised from one side for microhardness inspection of the welded plates. The description of the microhardness test is covered in the succeeding subsection.

### 2.3. Microstructure Examination

The samples had been prepared through the standard polishing procedure, which comprises polishing with SiC abrasive papers (having grit sizes ranging from 220–1000) abided by disc polishing for the mirror-like surface. The polished pieces were etched with 2% Nital (96% CH_3_OH + 4% HNO_3_) before being examined through microscopy (Olympus GX-51, Shinjuku, Japan) to uncover the morphology of welded steel plates.

### 2.4. Microhardness Test

The testing pieces have been shaped in compliance with the ASTM E384–99e1 requirements and assessed using Vickers microhardness equipment (Mitutoyo, made in Takatsu-ku, Japan). The dents on the testing pieces had been created with a 0.5 kgf load and a dwell duration of 15 s. At a span of 2 mm, the indents were made from the left-hand side to the right-hand side by covering the areas of base metal followed by weld bead and base metal (Figure 5).

## 3. Results and Discussion

### 3.1. Surface Morphology

The surface morphology of the welded joints was investigated using a 200× inverted microscope. This evaluation was carried out throughout the mid-thickness of the welded samples in order to explore morphological features in three regions, namely the BM, the HAZ, and the WZ. The surface morphology of such regions does have a significant impact over the structural attributes of the weldments. The surface morphology of the BM is shown in Figure 6. The essence of two different phases has been readily seen in it. The lighter region, which appears to be plentiful, indicates the prevalence of the ferritic phase, meanwhile, the darker zone indicates the prevalence of the pearlitic phase. This steel’s surface morphology in its original conditions is characteristic of the production technique, and it is analogous to the surface morphology of the exact metal documented in a few publications [43,44,45].

The surface morphologies of the weld junctions, namely BM-HAZ and HAZ-WZ, have been depicted in Figure 7a,b, correspondingly. Figure 7c exhibits the surface morphology of the WZ. These illustrations describe the particle refining phase from BM through HAZ to WZ, wherein particles have been turned from coarse-sized to fine-sized. The surface morphology variations in BM-HAZ (Figure 7a) reveal the emergence of pearlite structure, with the sample exhibiting a finer yet tightly packed pearlite lamella. The occurrence of the darker patch in HAZ verifies the prevalence of tiny pearlitic arrangement. Advancing from the HAZ to the WZ (in Figure 7b), the surface morphology darkens considerably, yet the acicular arrangement persists, suggesting the emergence of the bainite arrangement in the HAZ nearby the WZ. The surface morphology in Figure 7c is exceedingly finer and has a tree-like appearance, which demonstrates the prevalence of dendrites in the WZ. Dendrite formation, as documented by various academics, is notably prevalent in fusion-based weldments owing to the speedy cooling rate [46,47,48,49].

Post-heat treatment, the appearance of bainite (designated by the letter B) in a dark hue and ferrite (signified by the letter F) in a light hue can be seen in Figure 8a. Very fine dissemination of metal carbides, e.g., M_3_C, M_2_C, M_7_C_3_, and M_23_C_6_ could also be ascertained in the morphology, although their dissemination is difficult to recognize due to their narrow breadth. At this point, M stands for Fe, Cr, Mo, or a combination of such metals, whilst C stands for carbon. The morphology of BM and HAZ differs noticeably in Figure 8b. The microscopy of the base metal reveals the presence of bainite, meanwhile HAZ micrographs exhibit a combination of finer and coarser grains. These grains have the appearance of an acicular bainitic arrangement. The discovered micrographs are comparable to the Cr-Mo steel micrographs documented by Ahmed et al. [50], Das et al. [45], Yoo et al. [51], and Juliermes et al. [52] in the earlier investigations on Cr-Mo steel morphology. They identified the existence of bainite as well as different carbide precipitates.

The base metal was quenched and tempered for this analysis. Quenching is often performed by abruptly cooling the metal from its Ms point, culminating in uneven flakes as a consequence of the improper cooling speed. If cooling speeds are substantial, a martensitic arrangement formation occurs with a lathy look. While tempering, the flakes have enough chance to restructure themselves, causing a reduction in hardness. In repercussion, the lathy flakes take on circular forms. The WZ passes through an analogous quenching phase owing to the substantially lesser volume of heated liquid substance and the reduced neighboring temperatures. As a consequence of the rapid re-crystallization time, the WZ is exposed to an air-cooling action, culminating in erratic hard flake arrangements, such as an untampered martensitic phase [53,54,55]. Since the chosen element is air-hardened steel, the molten phase solidifies quickly owing to the faster cooling speed, culminating in the growth of martensitic arrangement. Once the PWHT is performed, the deformed BCC atomic structure, also known as the BCT (body-centered tetragonal) mechanism, has adequate opportunity to organize itself at predetermined locations. As a result, a tempering effect is evident. PWHT allowed the welded joints adequate space to recrystallize, culminating in the production of tempered martensite in WZ, as seen in Figure 8c. This work is also corroborated by the observations of Vora and Badheka [56,57], who investigated the morphology of A-TIG welded RAFM steel weldments and noticed martensite arrangement in WZ.

A closer look at the HAZ microscopy revealed two separate regions: the partially refined zone (PRZ) and the refined zone (RZ). These regions are depicted in Figure 9a. The PRZ strongly demonstrates a reduction in bainitic and ferritic phase thicknesses from BM to HAZ. In Figure 8b, the morphology of HAZ and WZ differs, as the grains in WZ look finer and more erratic than the grains in HAZ. This pattern explains the presence of martensite in WZ [38,39,46]. The development of such lamellae by an uneven distribution of grains during nucleation is caused by an insufficient solidification duration. Once the PWHT is introduced to weld joints, nevertheless, the lamellae coarsened somewhat, owing to limited restructuring [49,50,58].

### 3.2. Microhardness

Figure 10 depicts the variation in micro-hardness values throughout the welded plates. 

The hardness value for the as-welded RMD plate is highest in the weld zone, about 1588.6 HV, compared to 492.1 HV for the base metal. Similarly, the weld zone of heat-treated RMD plates has the maximum hardness value, about 227.3 HV, compared to the base metal, which has a hardness value of 164.1 HV. The cause behind such variations in the hardness values of as-welded and heat-treated RMD plates may be credited to the morphological changes in the materials, which can be vindicated by the literature mapping discussed as follows. As a result of the tempering, the basic material had rounded grains, resulting in decreased hardness as expected. The HAZ may have a combination of fine and coarse grains, resulting in a rising pattern in hardness levels. Moreover, there was a significant disparity in the weld zone’s hardness levels of as-welded RMD plates and heat-treated RMD plates. The emergence of harder and erratic martensitic formations may be responsible for the elevated hardness level of the weld zone in as-welded RMD plates. The extended lath width in martensitic clusters may account for the reduced hardness levels of the weld zone in heat-treated RMD plates. An additional conceivable reason could be the deposition of M23C6 and MX-type carbide precipitates as discussed in a couple of studies [57,59]. In addition to this, the measured hardness values reveal yet another corroboration of the observed surface morphologies as discussed in Section 3.1.

## 4. Conclusions

The study examines the effect of metal-cored filler wire on as-welded and heat-treated RMD welded low alloy steel plates. The optimized RMD welding variables A = 100 A, V = 13 V, and GFR = 21 L/m, were used for welding 06 mm thick ASTM A387-Gr.11-Cl.2 plates. This study examined and compared the materials conditions in terms of micro-hardness and surface morphology for as-welded and heat-treated conditions. Ferritic and pearlitic arrangements were seen during the surface morphology examination of the base metal prior to heat treatment. However, post-heat treatment, the existence of bainitic and ferritic arrangements was detected in the base metal. The surface morphology analysis of the weld zone of as-welded RMD samples revealed a very fine dendritic arrangement which is notably prevalent in fusion-based welding processes. On the other hand, the weld zone of heat-treated samples revealed the presence of tempered martensite. The micro-hardness value in the weld zone of the as-welded RMD sample was 1588.6 HV, which is very high. However, post-heat treatment, the micro-hardness values decreased drastically and the uniform variation in these values indicates the usefulness of the applied heat treatment cycle. Moreover, the obtained micro-hardness values further gave affirmation to the observed surface morphologies in as-welded and heat-treated RMD samples. In addition to this, this work may further be extended by characterizing the macroscopic details and mechanical attributes of the RMD welded samples.

## Figures and Tables

**Figure 1 materials-15-06661-f001:**
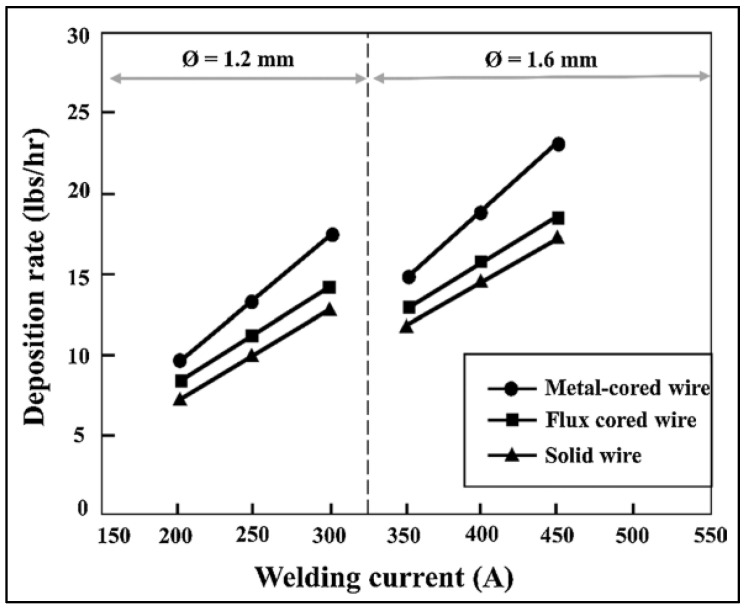
Deposition rates of filler wires: solid vs. flux-cored vs. metal cored [20].

**Figure 2 materials-15-06661-f002:**
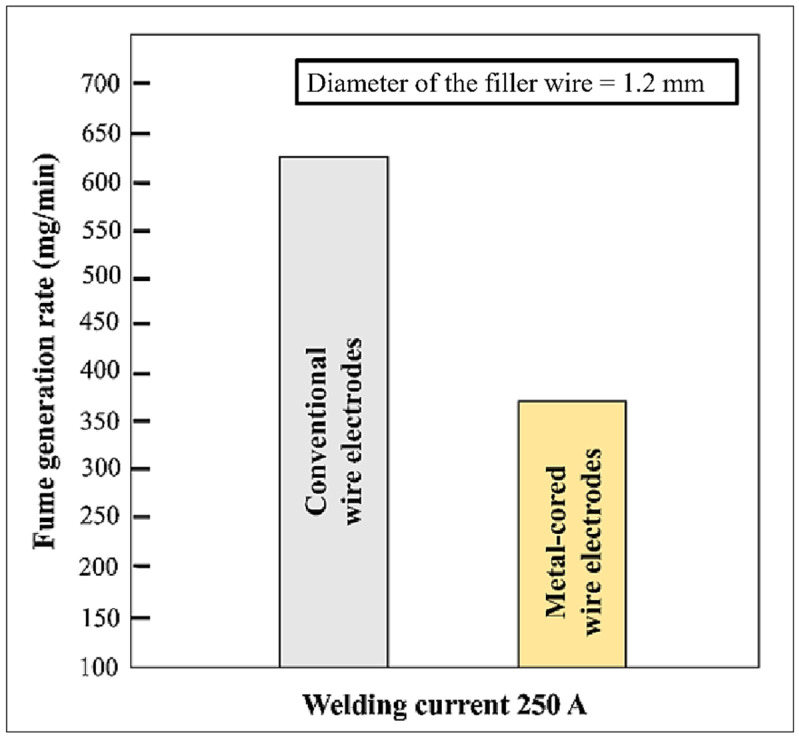
Fume generation: conventional filler wires vs. metal cored filler wires [20].

**Figure 3 materials-15-06661-f003:**
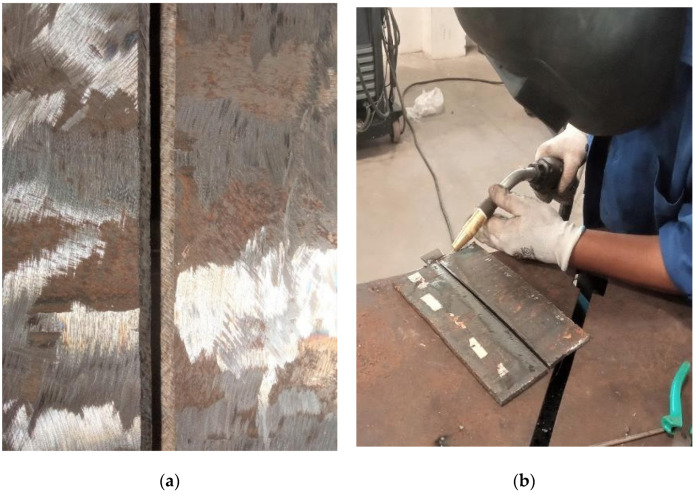
Grooving, tacking, and welding of steel plates. (**a**) groove. (**b**) tack. (**c**) welded plates.

**Figure 4 materials-15-06661-f004:**
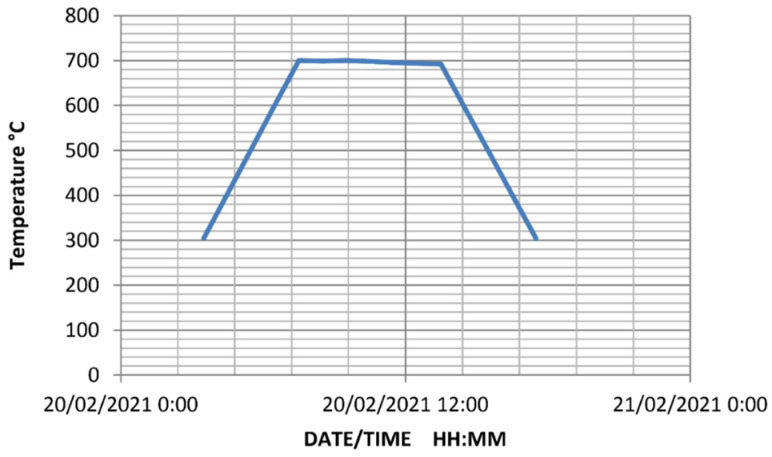
Thermal cycle of the heat treatment.

**Figure 5 materials-15-06661-f005:**
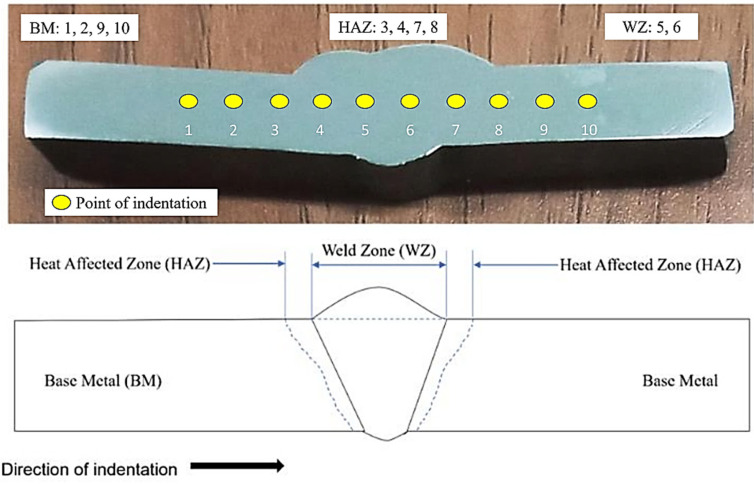
The direction and location of indentation for the micro-hardness assessment.

**Figure 6 materials-15-06661-f006:**
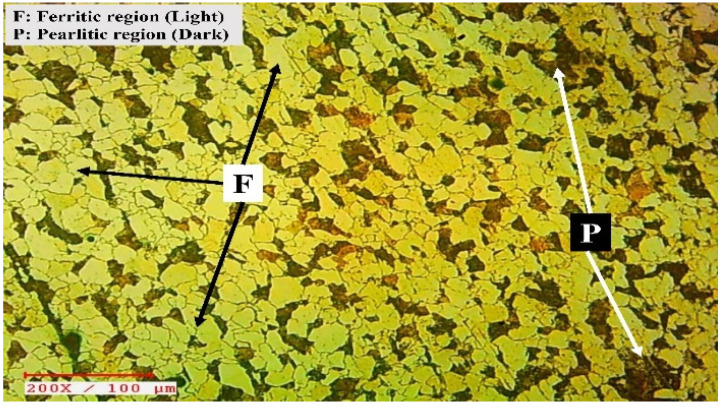
The surface morphology of base metal.

**Figure 7 materials-15-06661-f007:**
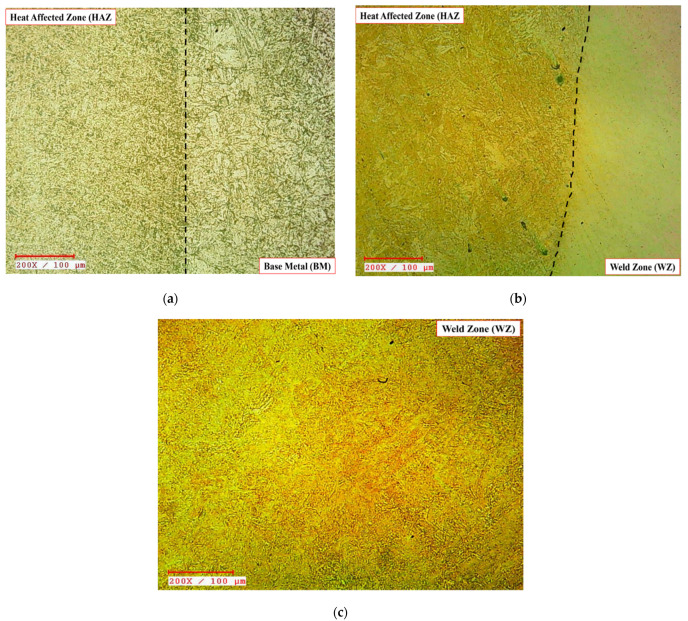
Surface morphology of the ASTM A387-Gr.11-Cl.2 weldments in various regions before heat treatment. (**a**) BM-HAZ interface. (**b**) HAZ-WZ interface. (**c**) WZ.

**Figure 8 materials-15-06661-f008:**
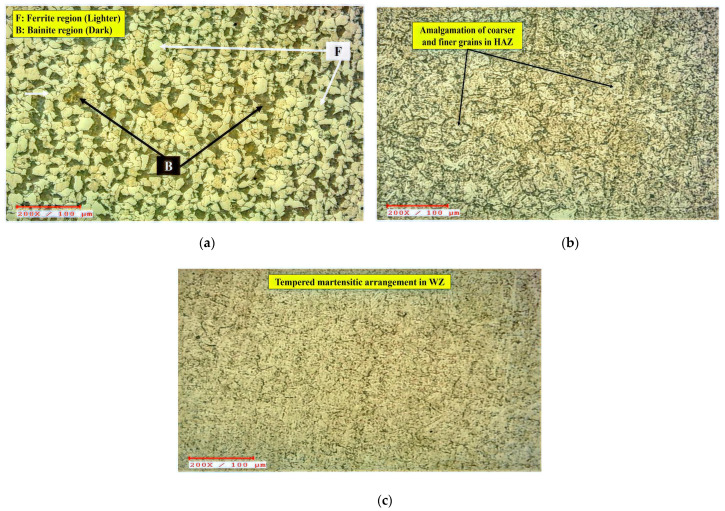
Surface morphology of the ASTM A387-Gr.11-Cl.2 weldments in various regions after heat treatment. (**a**) BM. (**b**) HAZ. (**c**) WZ.

**Figure 9 materials-15-06661-f009:**
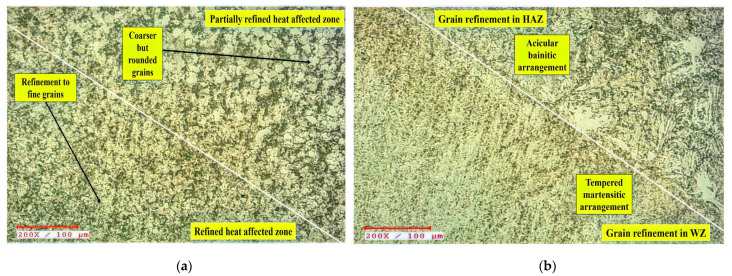
Surface morphological changes post-heat treatment in (**a**) HAZ and (**b**) HAZ-WZ interface.

**Figure 10 materials-15-06661-f010:**
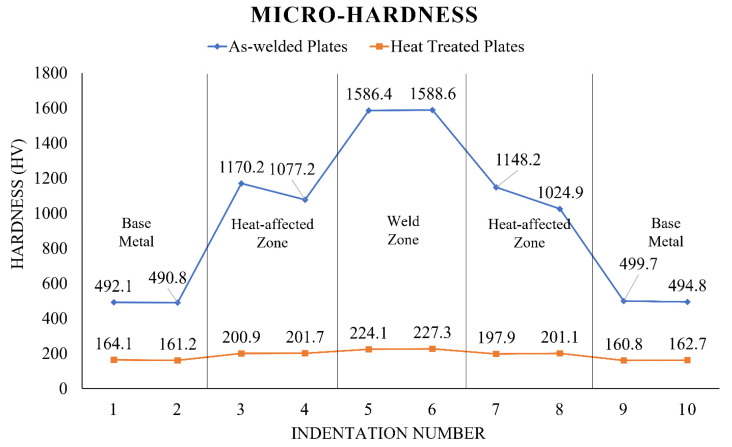
Values of micro-hardness test.

**Table 1 materials-15-06661-t001:** The chemical substance of MCFW and ASTM A378-Gr.11-Cl.2 steel.

	Cr	Mo	Mn	S	C	Si	P
MEGAFIL 237 M	2.3%	1.1%	1.0%	0.015%	0.07%	0.3%	0.015%
ASTM A387-Gr.11-Cl.2	1.00–1.50%	0.45–0.65%	0.40–0.65%	0.035%	0.05–0.17%	0.50–0.80%	0.035%

**Table 2 materials-15-06661-t002:** Particulars of experimental settings during welding.

Elements	MCFW	MCFW Diameter	Shielding Gas	Welding Position and Progression
Values	MEGAFIL 237 M	1.2 mm	90% Ar + 10% CO_2_	3G (Vertical) Downhill
Elements	GFR	Voltage	Current	Wire stick-out
Values	21 L/min	13 V	100 A	3–5 mm

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
