# Peer review of "Effect of Metal-Cored Filler Wire on Surface Morphology and Micro-Hardness of Regulated Metal Deposition Welded ASTM A387-Gr.11-Cl.2 Steel Plates"

_materials, 2022, doi:10.3390/ma15196661_

Round 1
Reviewer 1 Report
I can recommend the publication of this manuscript after a minor revision.
Line 113, fig. 2, explains the abbreviation used in the figure.
Lines 274-276. Is it possible to specify a percent on the surface? Can you specify some associated dimensions about the “acicular bainitic arrangement”?
Lines 291-293 – Insert more details. Explain the corresponding mechanism involved in the process described in line 292.
Lines 307-308, 309-310. Can you specify some technical values?
Line 313. Is it possible to explain this process in detail? Can you specify a reference?
Line 323. Improve the resolution/quality of fig. 9. Also, insert units for the vertical axis.
Line 369: minor mistake – delete “Please add:”
Specify the limits of this study. State in more detail the respective advantages and disadvantages.
Even though the work is relevant to the journal's scope, i.e., Materials, I do not find even a single article published in the journal in the list of references.
Authors may consider citing the following references:
[1] Ramesh Singh, Designing Weldments, Wiley, 2022.
[2] Ş. Ţălu, Micro and nanoscale characterization of three-dimensional surfaces. Basics and applications. Napoca Star Publishing House, Cluj-Napoca, Romania, 2015.
This paper can be published after the mentioned revisions.
Author Response
We would like to thank the anonymous reviewers for their careful reading of our manuscript and for coming up with several insightful comments and suggestions. Our response to the comments of each reviewer has been written below and also implemented in the revised manuscript.
******************************************************************************
Reviewer 1: Comments and Suggestions for Authors
I can recommend the publication of this manuscript after a minor revision.
- Line 113, fig. 2, explains the abbreviation used in the figure.
Response: As per the reviewer’s request, the abbreviation in Fig. 2 has been explained and incorporated in the revised manuscript.
- Lines 274-276. Is it possible to specify a percent on the surface? Can you specify some associated dimensions about the “acicular bainitic arrangement”?
Response: The authors agree with the reviewer’s comment and with due respect consider it for the next phase of the study related to the grain morphology of the materials. In that, the percentage of different microstructure arrangements and their dimensions along with the mechanical properties like the tensile test, impact test, etc., have been planned. Besides this, it is also well-known that grain morphology is an advanced field and due to its humongous depth, it becomes a separate area of research. Here, the authors' work is completely focused on examining the effect of metal cored filler wire on ASTM A387 Gr. 11 Cl2 steel during RMD welding and its mechanical characterization by evaluating the microhardness of the weldments. When the material’s properties come into the picture, it becomes obvious to explore the material's microstructure (which has been done in this study through optical microscopy).
- Lines 291-293 – Insert more details. Explain the corresponding mechanism involved in the process described in line 292.
Response: The authors have incorporated the suggestion in the revised manuscript.
- Lines 307-308, 309-310. Can you specify some technical values?
Response: Here, the authors mean the thickness as the shape and size of the grains. The grain size has not been measured in this study. However, under an optical microscope, the difference in size and shape can be easily seen in different zones of the weldments. As mentioned in response to suggestion number 2, the grain size measurement along with the exploration of various mechanical properties have been planned in the next phase of the study.
- Line 313. Is it possible to explain this process in detail? Can you specify a reference?
Response: As per the reviewer’s request, the references are cited and highlighted in the revised manuscript.
- Line 323. Improve the resolution/quality of fig. 9. Also, insert units for the vertical axis.
Response: As per the given suggestion, the authors have edited Fig. 9 and incorporated the modification in the revised file.
- Line 369: minor mistake – delete “Please add:”
Response: The suggested words have been removed in the revised file.
- Specify the limits of this study. State in more detail the respective advantages and disadvantages.
Response: Authors are grateful to the reviewers for this comment. In view of this point, the authors would like to state that the relative advantages and disadvantages of the metal cored filler wire (MCFW) with conventional filler wires have been discussed in the introduction section. Also, the authors have planned for a comparative study with MCFW using other conventional techniques for examining the mechanical characteristics.
- Even though the work is relevant to the journal's scope, i.e., Materials, I do not find even a single article published in the journal in the list of references. Authors may consider citing the following references:
[1] Ramesh Singh, Designing Weldments, Wiley, 2022.
[2] Ş. Ţălu, Micro and nanoscale characterization of three-dimensional surfaces. Basics and applications. Napoca Star Publishing House, Cluj-Napoca, Romania, 2015.
Response: The suggested references have been incorporated in the revised file.
Reviewer 2 Report
Dear authors, the manuscript ‘Effect of metal-cored filler wire on surface morphology and micro-hardness of regulated metal deposition welded ASTM A387-Gr.11-Cl.2 steel plates’, Manuscript ID: materials-1910532, have some weak issues that must be improved, sa follows:
1. Considering the “introduction’ section, it contains many valuable information to put the reader in a good requirements of the results presented. However, the critical review was found only in lines 143-146. Motivation is clear but please try to emphasize the lack of knowledge that are improved in the manuscript.
2. The selection of two steel plates, MEGAFIL 237 M and ASTM A387-Gr.11-Cl.2, were not justified. Why those were chosen?
3. More details about the surface morphology with a Olympus GX-51 microscopy should be added. What were the parameters when measuring?
4. Concluding to the previous comment, any information about the measurement uncertainty, noise or, generally, errors, would be also required. Please look for some examples for surface topography measurements:
(1) https://doi.org/10.1088/2051-672X/3/3/035004
(2) https://doi.org/10.3390/ma14020333
(3) https://doi.org/10.1088/0957-0233/23/3/035008
5. Even one, general (main) conclusion should be raised. Currently, in presented form, the conclusions are too detailed and connot put the reader into on way (idea).
6. The References formatting should be unified, according to the journal template requirements.
Concluding, the proposed manuscript is interesting. However, has some weaknesses and, at least in the current form, is not suitable for publication in a quality journal as the Materials is.
The manuscript must be improved significantly before any further processing, if allowed.
Author Response
We would like to thank the anonymous reviewers for their careful reading of our manuscript and for coming up with several insightful comments and suggestions. Our response to the comments of each reviewer has been written below and also implemented in the revised manuscript.
******************************************************************************
Reviewer 2: Dear authors, the manuscript ‘Effect of metal-cored filler wire on surface morphology and micro-hardness of regulated metal deposition welded ASTM A387-Gr.11-Cl.2 steel plates’, Manuscript ID: materials-1910532, have some weak issues that must be improved, as follows:
- Considering the “introduction’ section, it contains many valuable information to put the reader in a good requirements of the results presented. However, the critical review was found only in lines 143-146. Motivation is clear but please try to emphasize the lack of knowledge that are improved in the manuscript.
Response: Authors are thankful to the reviewer for his/her valuable suggestion. For any reader, the introduction of the current manuscript may seem inappropriate from a single variable point of view. However, a process optimization was accomplished for obtaining a single set of welding variables. In addition to this, the authors would like to inform the reviewer that this work is an extended version of the ongoing study which has been previously published in a reputed journal. The link to the published article is as follows:
https://www.tandfonline.com/doi/abs/10.1080/10426914.2021.1906897
In published work, the RMD welding variables were optimized for welding performance characteristics namely heat affected zone, depth of penetration, bead height, and bead width. Therefore, in this work, the authors have directly used the optimized data set as welding current (A) = 100 A, Voltage (V) = 13 V, and GFR = 21 L/min.
- The selection of two steel plates, MEGAFIL 237 M and ASTM A387-Gr.11-Cl.2, were not justified. Why those were chosen?
Response: The authors are grateful to the reviewer for this valuation remark. Here, the authors would like to state that compatibility plays a significant role in filler wire selection. The filler wire must be compatible with the base metal's chemical composition. By keeping this factor in mind, the selection of MEGAFIL 237 M filler wire was done. The chemical composition of the base metal and the filler wire has been presented in Table 1.
In addition to this, the authors would like to state that the work presented here is a continuation of the previous work published in a reputed journal. The link to the published article is as follows:
https://www.tandfonline.com/doi/abs/10.1080/10426914.2021.1906897
Meanwhile, the authors would like to add that ASTM A387-Gr.11-Cl.2 is a structural material that is quite regularly used in various applications like pressure vessels, oil and gas pipelines, chemical industries, and many others. So far, the welding of this material has been accomplished by conventional welding processes using conventional filler wires. Conventional welding processes lag in travel speed and others have the issue of spattering. On the other hand, solid and flux-cored filler wires have the issue of environmental contamination. Hence, an attempt has been made by this team to evaluate the mechanical and metallurgical aspects of implementing metal cored filler wires for ASTM A387-Gr.11-Cl.2 using RMD welding, which is a fast, self-controlled, and spatter-free technique.
- More details about the surface morphology with a Olympus GX-51 microscopy should be added. What were the parameters when measuring?
Response: For microstructural examination, the authors have taken care of the standards of sample preparation, magnification, material’s surface, location, and orientation. The first step was the careful selection of a small sample of the material to undergo microstructure analysis with consideration given to location and orientation. This step was followed by sectioning, mounting, grinding, polishing, and etching to reveal accurate microstructure and content.
- Concluding to the previous comment, any information about the measurement uncertainty, noise or, generally, errors, would be also required. Please look for some examples for surface topography measurements:
(1) https://doi.org/10.1088/2051-672X/3/3/035004
(2) https://doi.org/10.3390/ma14020333
(3) https://doi.org/10.1088/0957-0233/23/3/035008
Response: The authors are thankful to the reviewer for sharing the papers related to the microstructural evaluations. The authors would like to state that most of the guidelines were followed during the sample’s examination. However, the authors assure that for future studies, which have been already planned for this project, all these guidelines will be incorporated into the manuscript.
- Even one, general (main) conclusion should be raised. Currently, in presented form, the conclusions are too detailed and connot put the reader into on way (idea).
Response: As per the reviewer’s request, the conclusion has been revised. The authors have incorporated and highlighted the modified conclusion in the revised manuscript.
- The References formatting should be unified, according to the journal template requirements.
Concluding, the proposed manuscript is interesting. However, has some weaknesses and, at least in the current form, is not suitable for publication in a quality journal as the Materials is.
The manuscript must be improved significantly before any further processing, if allowed.
Response: The authors have implemented and highlighted the given suggestion in the revised manuscript.
Reviewer 3 Report
This paper presents an analysis of the surface morphology and microhardness of steel Gr.11-Cl.2 that was subjected to a welding process. The paper may be of some interest to the scientific community through the topic covered. The authors can consider the following aspects:
- The abstract must be rewritten, considering the fact that it presents too much general information, as it is necessary to present more technical data related to the results obtained in the research;
- The introduction should be improved in terms of content because a lot of general data related to the welding process is presented without emphasizing the analysis of previous research related to the topic addressed. It is not understood whether Figure 1 and Figure 2 are the authors' own achievements or are taken from the specialized literature;
-Choosing a single set of welding parameters does not represent research, and, thus, the whole paper has more the structure of a technical report;
- It is required to present the heat treatment diagram because otherwise the parameters are not very clear. For example, holding time is also very important;
- Applying a heating up to 700 C and a slow cooling I don't think can cause a martensitic structure to be obtained;
- It is very important that the metallographic structures shown are to the same scale. For example, in Figure 6 the HAZ is shown differently in a and b respectively;
- The research methodology is difficult to understand considering that no optimization is obtained but only some general results;
- A presentation of macroscopic images of the welded samples and after the applied thermal treatment is required;
- it is necessary to clearly show on the analyzed sample the position of the points where the microhardness was measured;
- The discussion part must be much developed in order to be able to highlight the novelty brought by the research presented in the paper in relation to other research in the field. In this form, it is not possible to identify the novelty of the research compared to previous research;
- Conclusions should be more concrete and future research directions should be presented.
Author Response
We would like to thank the anonymous reviewers for their careful reading of our manuscript and for coming up with several insightful comments and suggestions. Our response to the comments of each reviewer has been written below and also implemented in the revised manuscript.
******************************************************************************
Reviewer 3: This paper presents an analysis of the surface morphology and microhardness of steel Gr.11-Cl.2 that was subjected to a welding process. The paper may be of some interest to the scientific community through the topic covered. The authors can consider the following aspects:
# The abstract must be rewritten, considering the fact that it presents too much general information, as it is necessary to present more technical data related to the results obtained in the research
Response: As per the reviewer’s suggestion, the authors have rewritten the abstract by omitting the general information.
# The introduction should be improved in terms of content because a lot of general data related to the welding process is presented without emphasizing the analysis of previous research related to the topic addressed. It is not understood whether Figure 1 and Figure 2 are the authors' own achievements or are taken from the specialized literature;
Response: Authors are thankful for the reviewer’s comments and would like to state that the application of metal cored filler wires in regulated metal deposition (RMD) welding makes an interesting combination for researchers. Therefore, the authors have included almost all research in the literature related to RMD welding and the application of MEGAFILL 273 M filler wire. Additionally, the authors would like to bring this matter to the reviewer’s attention that the RMD welding has been patented by Miller Mfg. Ltd. Hence, there is a scarcity of literature related to RMD welding in the public domain, which is also one of the motivations behind this study.
Figure 1 and Figure 2 have been taken from the various available literature to justify the environmental aspects of the metal-cored filler wire. Also, the authors have cited the reference for these figures in the revised manuscript.
#Choosing a single set of welding parameters does not represent research, and, thus, the whole paper has more the structure of a technical report;
Response: Authors are thankful to the reviewer for his/her valuable suggestion. For any reader, the introduction of the current manuscript may seem inappropriate from a single variable point of view. However, a process optimization was accomplished for obtaining a single set of welding variables. In addition to this, the authors would like to inform the reviewer that this work is an extended version of the ongoing study which has been previously published in a reputed journal. The link to the published article is as follows:
https://www.tandfonline.com/doi/abs/10.1080/10426914.2021.1906897
In published work, the RMD welding variables were optimized for welding performance characteristics namely heat affected zone, depth of penetration, bead height, and bead width. Therefore, in this work, the authors have directly used the optimized data set as welding current (A) = 100 A, Voltage (V) = 13 V, and GFR = 21 L/min.
#It is required to present the heat treatment diagram because otherwise the parameters are not very clear. For example, holding time is also very important;
Response: The authors are thankful to the reviewer for this valuable comment The given suggestions have been implemented and highlighted in the revised manuscript.
# Applying a heating up to 700 C and a slow cooling I don't think can cause a martensitic structure to be obtained;
Response: Authors agree with the reviewer’s comment that the slow cooling doesn’t cause the formation of martensitic structure in the materials. But here, the authors would like to bring this point to the reviewer’s attention that the heat treatment cycle has been applied on RMD welded samples. RMD welding is a low heat input process. In the low heat input process, the cooling rate in the weld zone and its surroundings is quite high, leading to the formation of fine grains in the weld zone [1,2]. In this study, RMD welding is employed for welding steel plates. During welding (before the application of heat treatment), the cooling rate is usually high in the weld zone due to this it is very likely to obtain the martensitic transformation in the weld zone. Also, the obtained microhardness values in the study justify the claim.
[1] https://metall-mater-eng.com/index.php/home/article/view/342
[2] https://www.twi-global.com/technical-knowledge/faqs/what-is-the-heat-affected-zone
# It is very important that the metallographic structures shown are to the same scale. For example, in Figure 6 the HAZ is shown differently in a and b respectively;
Response: The authors are thankful to the reviewer for this valuable comment. The said image has been modified, incorporated, and highlighted in the revised manuscript.
# The research methodology is difficult to understand considering that no optimization is obtained but only some general results;
Response: Authors are thankful to the reviewer for his/her valuable suggestion. For any reader, the introduction of the current manuscript may seem inappropriate from an optimization point of view in his/her first reading. However, the authors would like to inform you that this work is an extended version of the ongoing study which has been previously published in a reputed journal. The link to the published article is as follows:
https://www.tandfonline.com/doi/abs/10.1080/10426914.2021.1906897
In published work, the RMD welding variables were optimized for welding performance characteristics namely heat affected zone, depth of penetration, bead height, and bead width. Therefore, in this work, the authors have directly used the optimized data set as welding current (A) = 100 A, Voltage (V) = 13 V, and GFR = 21 L/min.
#A presentation of macroscopic images of the welded samples and after the applied thermal treatment is required;
Response: Authors are thankful to the reviewer for this valuable suggestion as it gives another aspect of research in this area where macroscopic images of the weldments before and after heat treatment can be examined and compared. As mentioned in the previous comments that the study of RMD welding on steel plates incorporating metal cored filler wire is still going on and it consists of various phases where other aspects like mechanical characterization as well as macroscopic examination will be examined. However, here, for the sake of time being and keeping the duration of this project in mind as well as the length of the paper, it will be difficult to incorporate the said details in this study. Meanwhile, the authors are obliged to the reviewer for coming up with a suggestion and assure her/him to incorporate such details in the upcoming phase of the study.
#It is necessary to clearly show on the analyzed sample the position of the points where the microhardness was measured;
Response: The authors are thankful to the reviewer for this valuable comment. The given suggestion has been modified, incorporated, and highlighted in the revised manuscript.
#The discussion part must be much developed in order to be able to highlight the novelty brought by the research presented in the paper in relation to other research in the field. In this form, it is not possible to identify the novelty of the research compared to previous research;
Response: The result and discussion section include the discussion portion for each finding. Besides a discussion about the results obtained in the study, the section also includes the justification for the obtained result. For this, the authors have referred to several outcomes and cited them as a reference to justify the discussion and outcome of the study.
#Conclusions should be more concrete and future research directions should be presented;
Response: As per the reviewer’s comments, the future research direction of this work has been added to the conclusion section of the revised manuscript.
Round 2
Reviewer 2 Report
The manuscript was improved appropiately so, respectively, can be further processed by the Materials journal editorial office.
Author Response
Thank you very much for recommendation towards acceptance.
Reviewer 3 Report
The authors revised their manuscript according to my suggestions. Thus the manuscript can be accepted for publication.
Author Response

(The authors gave the same response as above.)
